# Lead-Free MDABCO-NH_4_I_3_ Perovskite Crystals Embedded in Electrospun Nanofibers

**DOI:** 10.3390/ma15238397

**Published:** 2022-11-25

**Authors:** Rosa M. F. Baptista, Gonçalo Moreira, Bruna Silva, João Oliveira, Bernardo Almeida, Cidália Castro, Pedro V. Rodrigues, Ana Machado, Michael Belsley, Etelvina de Matos Gomes

**Affiliations:** 1Centre of Physics of Minho and Porto Universities (CF-UM-UP), Campus de Gualtar, University of Minho, 4710-057 Braga, Portugal; 2Institute for Polymers and Composites, Campus de Azurém, University of Minho, 4800-058 Guimarães, Portugal

**Keywords:** organic lead-free perovskites, piezoelectric crystals, nanofiber composites, electrospinning, blue luminescence, functional organic materials

## Abstract

In this work, we introduce lead-free organic ferroelectric perovskite N-methyl-N′-diazabicyclo[2.2.2]octonium)–ammonium triiodide (MDABCO-NH_4_I_3_) nanocrystals embedded in three different polymer fibers fabricated by the electrospinning technique, as mechanical energy harvesters. Molecular ferroelectrics offer the advantage of structural diversity and tunability, easy fabrication, and mechanical flexibility. Organic–inorganic hybrid materials are new low-symmetry emerging materials that may be used as energy harvesters because of their piezoelectric or ferroelectric properties. Among these, ferroelectric metal-free perovskites are a class of recently discovered multifunctional materials. The doped nanofibers, which are very flexible and have a high Young modulus, behave as active piezoelectric energy harvesting sources that produce a piezoelectric voltage coefficient up to *g*_eff_ = 3.6 VmN^−1^ and show a blue intense luminescence band at 325 nm. In this work, the pyroelectric coefficient is reported for the MDABCO-NH_4_I_3_ perovskite inserted in electrospun fibers. At the ferroelectric–paraelectric phase transition, the embedded nanocrystals display a pyroelectric coefficient as high as 194 × 10^−6^ Cm^−2^k^−1^, within the same order of magnitude as that reported for the state-of-the-art bulk ferroelectric triglycine sulfate (TGS). The perovskite nanocrystals embedded into the polymer fibers remain stable in their piezoelectric output response, and no degradation is caused by oxidation, making the piezoelectric perovskite nanofibers suitable to be used as flexible energy harvesters.

## 1. Introduction

Mechanical energy harvesting at low frequencies from materials that are environmentally friendly and scavenging energy from multiple sources, for example, human body movements, are at the forefront of research [1,2].

Ferroelectrics are inherently piezoelectric and pyroelectric materials; that is, they are able to produce an intrinsic electrical potential difference in response to an applied force (or originate a mechanical movement due to an applied electric field) and a temperature gradient, respectively.

Valasek discovered the first ferroelectric, Rochelle salt or potassium sodium tartrate tetrahydrate [KNaC_4_H_4_O_6_] (4H_2_O) in 1920 and was, in fact, the first semiorganic molecular ferroelectric crystal that is also nontoxic [3,4].

Among ferroelectrics, the inorganic perovskites (formula ABX_3_ (A, B = metal cations, X = anion; usually an oxide)) are a well-known family of solid-state inorganic compounds finding application in capacitors, sensors, actuators, etc. The best known are metal oxides such as strontium, barium, or lead titanate (SrTiO_3_, BaTiO_3_, PbTiO_3_, respectively), and their solid solutions such as Pb(Zr,Ti)O_3_ (PZT), niobates such as PZN (PbZn_1/3_Nb_2/3_O_3_) (PZN), (PbMg_1/3_Nb_2/3_O_3_) (PMN), and lithium niobate LiNbO_3_). These materials have, until now, been used by several industries largely because of their functional properties, their combining ferroelectricity with nonlinear optical and electro-optic effects, as well as their multiferroicity [5,6].

So far, the commercially available piezoelectrics are dominated by inorganic perovskites, namely PZT-based materials, and polymers such as polyvinylidene difluoride (PVDF) and its modifications such as PVDF-TrF [7,8]. However, because of lead toxicity, lead-based ferroelectrics are presently a serious environmental hazard. These concerns originated active research on substituting those perovskite-type materials, one ion A or X, with a molecular building unit [9,10].

Hybrid organic–inorganic perovskites (HOIPs) are a recent class of ferroelectric crystalline materials for optoelectronic applications, which are competitive with inorganic perovskites. These semiorganic ferroelectrics possess many advantages when compared with inorganic ones. For example, they can be synthesized at room temperature, they are more flexible and with lower weight than their inorganic counterparts, and they have remarkable structural variability resulting in high tunable properties. Therefore, they became an attractive research topic for their application as piezoelectric and pyroelectric materials that replace inorganic materials [11,12,13,14]. Importantly, highly efficient solar cells have been demonstrated using methylammonium lead halide perovskites, which enabled the search for lead-free perovskites. Lead-free HOIPs are a recently discovered and highly promising family of perovskites [15,16,17,18,19,20].

A lead-free organic–inorganic perovskite recently discovered is (N-methyl-N′-diazabicyclo[2.2.2]octonium)–ammonium triiodide (MDABCO-NH_4_I_3_), which has a spontaneous polarization of 22 µC/cm^2^, close to that of barium titanate (which is around 26 µC/cm^2^), a high phase transition temperature at 448 K, and several polarization directions. It displays attractive properties for applications in flexible optoelectronic devices [21,22].

The fabrication of structures at the nanoscale has been attracting an increased amount of attention because of their size-dependent properties. One-dimensional structures such as nanowires, nanotubes, and nanofibers are the smallest dimensional structures displaying new properties with potential applications in fields such as electronics, photonics, sensing, and energy harvesting.

Electrospinning is a well-established technique for forming micro- and nanoscale fibers with a large surface-to-volume ratio forming mats of several square centimeters area. Electrospun fiber mats are nanostructured multifunctional materials drawn from a precursor polymeric solution blended with functional nanoparticles under very strong static electric fields [23,24,25,26,27].

In addition, the nanofiber’s anisotropic shape and large surface area ratio contribute to an increase in their mechanical strength and flexibility. In this context, nanoscale ferroelectrics with perovskite structure is a promising research area [11,28].

One application of functional electrospun fibers is in the harvesting of electrical nanoenergy at low frequencies through the piezoelectric effect because of the polarization induced by the material deformation [29]. Piezoelectric nanogenerators, usually called PENGs, show potential for powering low-power devices. An example of the use of a semiorganic perovskite as a PENG was reported for the methylammonium lead iodide (CH_3_NH_3_PbI_3_) incorporated in PVDF polymer fibers made by electrospinning; an output voltage of approximately 220 mV at 4 Hz, under an applied force of approximately 7.5 N, a maximum output power of 0.8 mW/m^2^ was generated [30].

In this manuscript, MDABCO-NH_4_I_3_ perovskite embedded into electrospun nanofibers is capable of acting as lead-free piezoelectric (PENG) nanogenerators for effective mechanical energy harvesting. In particular, for poly(vinyl chloride) (PVC) polymer, an instantaneous output power density of 2020 μWm^−2^ is delivered under the application of a mechanical periodical force. The pyroelectric coefficient of a polycrystalline MDABCO-NH_4_I_3_ in electrospun fibers has a similar order of magnitude to that displayed by hybrid ferroelectric triglycine sulfate (TGS).

## 2. Experimental Section

### 2.1. Materials and Nanofibers Preparation

MDABCO-NH_4_I_3_ was synthesized following the synthetic procedure reported by Yu-Meng You and Ren-Gen Xiong [21]. The precursor (MDABCO)I was synthesized, as reported by Kreuer et al. [31]. The MDABCO-NH_4_I_3_ crystals grown were ground in a mortar and sieved to a size smaller than 40 µm.

All chemicals and solvents were purchased from Sigma-Aldrich (Schenlldorf, Germany) and used as received. Poly (methyl methacrylate) (PMMA, Mw 120,000) was purchased from Alpha-Aesar (Kandel, Germany). Polyamide 66 (PA66) and Poly(vinyl chloride) (PVC), high molecular weight, a density of 1.40 g/mL), were purchased from BDH Chemicals (Poole, UK) and Janssen (Beerse, Belgium), respectively. The 10% polymer solution (*w*/*v*) of PMMA was prepared by dissolving the powder in a dichloromethane (DCM)/*N*,*N*-dimethylformamide (DMF) solvent blend system (80:20, *v*/*v*), with vigorous stirring (400–600 rpm) at room temperature. The 10% (*w*/*v*) of polymer solution of PA66 was prepared by dissolving the polymer in 5 mL of 1,1,1,3,3,3-hexafluoro-2-propanol (HFP) with vigorous stirring (400–600 rpm) at room temperature. Then, a 10% precursor electrospinning solution of PVC was prepared by dissolving the pellets in 5 mL of the tetrahydrofuran (THF)/DMF (50:50, *v*/*v*) solvent blend system. After complete dissolution, 0.1 g of MDABCO-NH_4_I_3_ was ground and incorporated in small portions in a 1:5 weight ratio, and the resulting solution was sonicated for 10 min and stirred for several hours under ambient conditions before the electrospinning process, shown in Figure 1.

The precursor solution was loaded into a 5 mL syringe with its needle (0.5 mm outer diameter and 0.232 mm inner diameter) connected to the anode of a high-voltage power supply (Spellmann CZE2000, Bochum, Germany). The nanofibers were produced by a conventional electrospinning technique, previously described in [26,32], with a configuration that tends to produce oriented fiber mats. Briefly, the equipment used has four common components: a high-voltage power supply, a precision syringe pump, a syringe fitted with a metal needle (spinneret), and a drum collector (connected to a motor speed controller). The power supply is connected to both the spinneret and the drum collector. The polymer solution is extruded through the spinneret at a constant flow rate controlled by the syringe pump. An aluminum foil is attached to the collector in order to collect the prepared fibers. Our electrospinning apparatus has a vertical geometry.

Polymer nanofibers with embedded MDABCO-NH_4_I_3_ perovskite using three different polymers were fabricated. Solutions with pure PVC, PMMA, and PA66 polymers were also electrospun and taken as a reference. For the electrospinning of MDABCO-NH_4_I_3_ containing polymers and reference solutions, a voltage of 18 kV was applied between the tip and collector. The flow rate of the solution and the needle-to-collector distance were kept at 0.18–0.30 mL/h and 12 cm, respectively.

The MDABCO-NH_4_I_3_ crystals are not stable in the open air at room temperature. When the crystals are exposed to air, the perovskite oxidizes, as shown in Figure 2a,b. The iodide ions slowly oxidize. The product is molecular iodine, I_2_, which darkens the crystals. Previously, to prepare precursor electrospinning solutions, several solvents were tested. In Figure 2c, it is possible to see that for solvents tetrahydrofuran (THF), methanol, ethanol, and acetone, the intense dark yellow color appears because of the degradation of the perovskite. Hexafluoroisopropanol (HFP) was chosen to prepare the electrospun polymer-doped solutions because the solutions remain stable, and the perovskite is protected from oxidation for long periods of time after preparation.

To make good use of the outstanding crystal properties, we found that, by embedding them into a polymer matrix, the perovskite crystals are protected from oxidation while keeping their optical, piezoelectric, and pyroelectric properties. The perovskite nanocrystals, when embedded into the fibers, are stable for more than half a year because the polymers function as shields protecting the perovskites from oxidation.

PMMA was chosen to prepare the hybrid matrix with MDABCO-NH_4_I_3_ organic perovskite because it is a biocompatible polymer. PA66, a polymer with a high melting point at 275–280 °C, was also chosen because it enables the measurement of the pyroelectric effect on the perovskite nanofibers near their Curie temperature. Finally, PVC polymer was chosen because of its nontoxicity, flexibility, strength, and high melting point around 220 °C. The electrospinning process was stable for all the polymers chosen, and the obtained fibers showed uniform surfaces and small diameters, demonstrating that there were no crystallites on their surface. The hybrid functional MDABCO-NH_4_I_3_@PVC, MDABCO-NH_4_I_3_@PA66, and MDABCO-NH_4_I_3_@PMMA nanofibers were further utilized for optical and dielectric characterization, as well as the exemplification of a piezoelectric voltage generator.

### 2.2. Scanning Electron Microscopy (SEM)

The morphology, size, and shape of MDABCO-NH_4_I_3_ perovskite nanofibers were studied using a Nova Nano SEM 200 scanning electron microscope (FEI Company, Hillsboro, OR, USA), operated at an accelerating voltage of 10 kV. Nanofibers were deposited on a silica surface previously covered with a thin film (10 nm thickness) of Au-Pd (80–20 weight %) using a high-resolution sputter cover, 208HR Cressington Company (Watford, UK), coupled to a Cressigton MTM-20 high-resolution thickness controller. The diameter range of the nanofibers was measured by SEM images using ImageJ 1.51n image analysis software (ImageJ2, NIH, https://imagej.nih.gov/ij/, 12 September 2022). The average diameter and diameter distribution were determined by measuring 80 random nanofibers from the SEM images, and the fiber diameter distributions fit to a log-normal function.

### 2.3. X-ray Diffraction and Raman Spectroscopy

The crystallinity and crystallographic orientation of MDABCO-NH_4_I_3_ inside the fibers were studied by X-ray diffraction. The diffraction pattern using θ–2θ scans was recorded on a Bruker D8 Discover (Bruker company, Billerica, MA, USA) with Cu-Kα radiation of wavelength 1.5406 Å.

Raman spectroscopy was performed on a LabRAM HR Evolution confocal Raman spectrometer (Horiba Scientific, France, Lille) using Horiba Scientific’s Labspec 6 Spectroscopy Suite software (LabSpec-Version 6) for instrument control, data acquisition, and processing. The Raman spectra were obtained using a laser excitation with wavelength 532 nm, at 0.1% laser intensity, with 30 s acquisition time in a spectral range between 50 and 3500 cm^−1^.

### 2.4. Mechanical Tests

Elastic modulus, stress at yield (at 0.2% offset), tensile strength, and strain at break (at 60% tensile strength) were measured using a universal tensile testing machine Zwick/Roell Z005 (ZwickRoell, Germany), following the ASTM D882–02 standard. Several 10 × 30 mm samples, with a gauge length of 16 mm, were tested alongside the oriented fiber direction under a crosshead velocity of 25 mm/min.

### 2.5. Optical Absorption and Photoluminescence

Optical absorption (OA) measurements on an MDABCO-NH_4_I_3_ solution were carried out using a Shimadzu UV-3101PC UV–Vis-NIR (Shimadzu Corporation, Kyoto, Japan) spectrophotometer. Photoluminescence spectra were recorded on a Fluorolog 3 spectrofluorimeter (HORIBA Jobin Yvon IBH Ltd., Glasgow, UK). For optical absorption measurements, a 3 mg/mL solution of MDABCO-NH_4_I_3_ was prepared in water. The sample was measured in a quartz cuvette with a 1 cm path length. Photoluminescence (PL) spectra were acquired using an excitation wavelength of 290 nm, with input and output slits fixed to provide a spectral resolution of 3 nm.

The same spectrophotometer equipped with an integrating sphere, Shimadzu ISR-240A (Shimadzu, Duisburg, Germany), and barium sulfate taken as reference, was used to measure the diffuse reflectance spectrum for the nanofiber array in the wavelength range of 250–800 nm with 1 nm step size. The energy of the band gap (E_g_) was determined using the Kubelka–Munk function given by [[hvF(R)]^1⁄2^] = α(hv − E_g_), where hv represents the energy of the incident photon, E_g_ corresponds to the energy of the bad gap, and F(R) is called the Kubelka–Munk function directly determined from the total reflectance coefficient of the material (R) through the equation F(R) = (1 − R)^2^/2R [33,34].

### 2.6. Dielectric Spectroscopy

The dielectric properties of the electrospun fibers with embedded MDABCO-NH_4_I_3_ inclusions were characterized by impedance spectroscopy, at temperatures of 300–460 K and in the frequency range of 20 Hz–3 MHz. The complex permittivity, written as ε = ε′ − iε″, where ε and ε″ are the real and imaginary parts, respectively, were calculated from the measured capacitance (C) and loss tangent (tan δ), using the equations:C = ε′ε_0_(A/d)   and    tan δ = ε″/ε′

Here A is the electric contact area, and d is the fiber mat thickness. To perform the measurements, the samples formed a parallel plate capacitor and were included in an LCR network. To form the capacitor, the aluminum foil used as the substrate to collect the fiber mats was the bottom electrode, while the top electrode was the base of cylindrical metal contact, with approximately 10^−2^ m diameter. A Wayne Kerr 6440A (Wayne kerr Electronics, London, UK) precision component analyzer was used together with a dedicated computer and software to acquire the data. Shielded test leads were employed to avoid parasitic impedances due to connecting cables. Temperature-dependent measurements were performed at a rate of 2 °C/min, using a Polymer Labs PL706 PID controller (Polymer Labs, Los Angeles, CA, USA) and furnace.

Pyroelectricity results from the temperature dependence of spontaneous polarization. By changing the temperature, an electric field originating from changes in intrinsic dipoles is compensated by the surface layer of free charges. The rate of change in the spontaneous polarization p = dP_s_/dT is the pyroelectric coefficient. The change in polarization was detected by measuring the pyroelectric current I = A (dPs/dT)(dT/dt) with a Keithley 617 electrometer (Keithley Instruments GmbH, Landsberg, Germany), where A is the electrode area and dT/dt is the rate of temperature change. The measurements were performed in a capacitor geometry under short-circuit conditions.

### 2.7. Fabrication of an MDABCO-NH_4_I_3_@PVC Piezoelectric Nanogenerator

A piezoelectric nanogenerator, fabricated using an MDABCO-NH_4_I_3_@PVC electrospun fiber mat as the active piezoelectric component, is described in Figure 3. The top and bottom electrodes (area 40 × 40 mm^2^) are high-purity copper thin plates. Thin copper wires were attached to the electrodes. The entire system was laminated with 1mm thick cork sheets to protect and facilitate the handling of the nanogenerator.

## 3. Results and Discussion

### 3.1. Electrospun Fibers

The perovskite polymer solutions obtained remain stable with no color change for several weeks, shown in Figure 4a. The electrospinning process is stable, with no current fluctuations and a steady flow of the polymer solution at the tip of the needle. The fabricated fibers are very flexible (inset), show no ‘beads’ or crystallites grown on their surface, have a white appearance and are flexible, Figure 4b.

### 3.2. Fibers Morphology and Crystallinity

Figure 5 shows scanning electron microscopy (SEM) images of the fibers prepared with different polymers with embedded MDABCO-NH_4_I_3_ perovskite, along with the corresponding histograms of the diameter sizes. The diameter distributions are observed to follow a log-normal dependence, with average values from 200 to 605 nm.

X-ray diffraction patterns obtained from the doped fibers are shown in Figure 6b–d and compared to the correspondent pattern for the polycrystalline synthesized MDABCO-NH_4_I_3_ perovskite, Figure 6a, where all the Bragg peaks were indexed using the published crystal structure (CIF file 1836322) [21]. We conclude that the embedded perovskite is in its crystalline ferroelectric phase for all fibers and is randomly oriented inside the different polymer matrices. The crystallite size of the perovskites was evaluated for each polymer, from fitting with the Debye–Scherrer equation the two most intense Bragg reflections $\left(11\bar{1}\right)$ and $\left(200\right)$, see Appendix A–c. The average size varies between 62 and 83 nm, as indicated in Table 1.

### 3.3. Optical Absorption and Luminescence

The reflectance spectra of an MDABCO-NH_4_I_3_ pellet and MDABCO-NH_4_I_3_@PVC nanofibers show two absorption bands with the maximum at wavelengths of 297 nm and 365 nm and 298 nm and 367 nm, respectively, as shown in Figure 7. PVC electrospun fibers are highly transparent in all UV–Vis spectra. The energy band gap, E_g_, calculated for pellet crystals and nanofibers, from the intersection with the energy axis of a linear Kubelka–Munk function, are 4.760 eV and 4.824 eV, respectively, as indicated in the insets of Figure 7. These values are in excellent agreement with 4.950 eV, as previously reported in [35]. Similar reflectance spectra measured for MDABCO-NH_4_I_3_@PMMA and MDABCO-NH_4_I_3_@PA66 nanofibers are shown in Appendix A.

At high wavelengths (low photon energies), both free perovskite and electrospun fibers containing the perovskite are highly transparent, and the absorption becomes stronger at the band-gap energy. High transparency in the visible and near-infrared regions of the optical spectra is important for linear and nonlinear optical applications [36,37].

Figure 8 shows the OA of a water solution MDABCO-NH_4_I_3_, the emission of PL from the same solution, and the dissolved MDABCO-NH_4_I_3_@PVC fibers for excitation at 289 nm. For the MDABCO-NH_4_I_3_ water solution, the PL emission shows one intense band in the UV with a maximum at 325 nm, a slightly lower intense band in the blue with a maximum at 395 nm, and a redshifted band with a maximum at 645 nm. For nanofibers (the solvent used dissolves only the polymer and not the perovskite nanocrystals), an extremely intense PL band at 325 nm and a less intense redshifted band with a maximum at 645 nm are observed. It is remarkable that MDABCO-NH_4_I_3_ nanocrystals inside the fibers show intense solid-state UV and blue luminescence, which is reported in this study for the first time for MDABCO-NH_4_I_3_ perovskite nanocrystals.

### 3.4. Dielectric Measurements

The complex dielectric permittivity measured on a polycrystalline sample of MDABCO-NH_4_I_3_ (pellet), between 300 K and 470 K as a function of frequency, shows that the ferroelectric–paraelectric phase transition occurs at 440 K, shown in Figure 9a,b. As expected for a proper ferroelectric system, the real part of the permittivity increases with decreasing frequency, reaching the Curie transition temperature of 42,500 (at 20 Hz). Quite extraordinarily, this very high value of ε′, measured on a pellet, is 3000 times higher than that reported for an oriented single crystal, which was 14,068 at the same frequency of 20 Hz [21]. This indicates the high purity of our synthesized MDABCO-NH_4_I_3_ crystals.

The real and imaginary parts of the dielectric permittivity were also measured on an MDABCO-NH_4_I_3_@PA66 electrospun fiber mat in the same temperature and frequency range, as shown in Figure 10 for frequencies below 1000 Hz. The transition is perceptible at 462 K, a little above the temperature of 440 K measured for a polycrystalline perovskite (Figure 9). When ferroelectric nanostructures are under stress/strain, their ferroelectric transition temperature varies compared with bulk unstressed ones. The shift in transition temperature depends on the strain state of the crystal. For example, hydrostatic strain tends to decrease the transition temperature [38], while anisotropic strain states can strongly increase the transition temperature [39,40]. In our case, from the X-ray diffraction results of Figure 6, we observe a slight shift of the XRD peaks compared with the bulk, indicating the MDABCO-NH_4_I_3_ nanocrystals are under strain inside the fibers. This nanofiber-induced strain is anisotropic because of the high aspect ratio of the nanofibers and has increased the transition temperature compared with the bulk. As such, the observed increased transition temperature results from the fact that the nanocrystals are immersed in the polymer matrix, making it necessary to go higher in temperature for the dispersed nanocrystals to make the ferroelectric–paraelectric transition temperature. The permittivity results also indicate a diffuse character of the phase transition, widened as compared with the bulk, induced by the small size of the MDABCO-NH_4_I_3_ nanocrystals [41,42] embedded in the polymer matrix.

### 3.5. Pyroelectricity in Fibers

In this work, we report for the first time the measurement of the pyroelectric coefficient of polycrystalline MDABCO-NH_4_I_3_ perovskite (a pellet) and nanocrystals embedded in electrospun fibers of PA66, that is, MDABCO-NH_4_I_3_@PA66. Note that PA66 is the only polymer that allowed the pyroelectric measurement to be carried out since both PMMA and PVC melt before the perovskite phase-transition temperature is achieved. The measured coefficients, as a function of temperature, are shown in Figure 11 for a fiber mat.

Extraordinarily, the pyroelectric coefficient of the MDABCO-NH_4_I_3_ nanocrystals increases to a very high value of 194 × 10^−^^6^ Cm^−^^2^k^−^^1^ achieved at 483 K, which is slightly above the Curie transition temperature value obtained for measurements of dielectric permittivity. This indicates a diffuse phase transition, which is expected to occur for nanocrystals randomly oriented inside a polymer matrix. The pyroelectric coefficient value obtained is within the same order of magnitude as that reported for the state-of-the-art semiorganic ferroelectric triglycine sulfate (TGS) single crystal, reported being 306 × 10^−^^6^ Cm^−^^2^k^−^^1^ at the ferroelectric–paraelectric phase transition [43].

### 3.6. Piezoelectric Voltage and Effective Piezoelectric Coefficients in Fibers

The behavior of the electrospun nanofiber mats fabricated from MDABCO-NH_4_I_3_@PA66, MDABCO-NH_4_I_3_@PMMA, and MDABCO-NH_4_I_3_@PVC is now studied as a piezoelectric energy generator. The generated open-circuit voltage, Voc, and short-circuit currents, Isc, are shown in Figure 12 as functions of the external forces applied. In Appendix A, the piezoelectric current generated by an MDABCO-NH_4_I_3_@PA66 fiber mat is shown. There is a linear relationship between the output electric current generated and the applied external forces, as expected for a piezoelectric material. Appendix A shows the output voltage generated from the MDABCO-NH_4_I_3_@PVC nanofiber mat with reverse polarity. To analyze the reproducible behavior of the piezoelectric active nanofiber mat as a nanogenerator, a stability test was performed during a time interval of 4 h, uninterruptedly, under a periodical force applied with 2.7 N at a frequency of 3 Hz, Appendix A. The nanogenerator output voltage does not decrease over time. This is an important property indicating that the MDABCO-NH_4_I_3_ perovskite nanocrystals may be used to integrate future nanogenerators devices. For the MDABCO-NH_4_I_3_@PVC fiber mat, Appendix A shows the piezoelectric current generated during a time interval of 130 s under a periodical force applied with 4.5 N at a frequency of 3 Hz, as well as for frequencies between 1 Hz and 10 Hz.

The MDABCO-NH_4_I_3_@PVC nanofibers are very flexible and show a tensile strength and Young modulus of approximately 4.0 MPa and 58 MPa, respectively, as shown in Appendix A. Flexibility and high Young modulus are important characteristics for the performance of a fiber mat as a nanogenerator because they will increase its capacity to last for longer under an applied external force.

For a frequency of 3Hz and compression stress of 11 kPa, *V_oc_* reaches 16.5 V for the PENG formed by the MDABCO-NH_4_I_3_@PVC nanofiber mat. For MDABCO-NH_4_I_3_@PMMA and MDABCO-NH_4_I_3_@PA66, *V_oc_* reaches 6.1 V and 2.0 V, respectively. Low frequencies, such as 3Hz, are those that enable the generator to return to its original microscopic configuration before the next force is applied.

The charge generated by a piezoelectric mat, calculated from Q = ∫Idt (C), results in charge of 786 pC for the MDABCO-NH_4_I_3_@PVC mat considering a material response time of the order of 10^−3^ s and the maximum *I_sc_* obtained of 786 nA. For MDABCO-NH_4_I_3_@PMMA and MDABCO-NH_4_I_3_@PA66 fiber mats, the charges generated are 288 pC and 95 pC, respectively. We may now calculate the effective piezoelectric coefficient, given by *d_eff_* = Q/F (pCN^−1^), for the three nanofiber mats under a periodical force applied at 4.5 N. These are 175 pCN^−1^, 64 pCN^−1^, and 21 pCN^−1^, for MDABCO-NH_4_I_3_@PVC, MDABCO-NH_4_I_3_@PMMA, and MDABCO-NH_4_I_3_@PA66, respectively. The piezoelectric coefficient reported for a single MDABCO-NH_4_I_3_ crystal is *d*_33_ = 14 pCN^−1^ along the [1 1 1] direction of the crystal [21]. Therefore, the piezoelectric coefficient for the hybrid system formed by MDABCO-NH_4_I_3_ nanocrystals embedded in PVC fibers is one order of magnitude higher. Note that for this hybrid perovskite PENG, there is a small contribution from the piezoelectric polymer, which is piezoelectric (*d*_31_~1.5 pCN^−1^) [44]. The present result is consistent with the very high effective piezoelectric coefficient displayed by electrospun fibers incorporated with active organic piezoelectric materials, which has been previously reported for nonlinear optical organic crystal derivatives of nanocrystalline push–pull nitroaniline molecules and diphenylalanine dipeptides, when embedded in nano and microfibers fabricated by the electrospinning technique [45,46,47].

It is also important to calculate the peak power density, P = *I_sc_*/A (μWm^−2^) (A is the area of the electrode) delivered by the MDABCO-NH_4_I_3_@PVC nanofiber mat, which amounts to 1960 μWm^−2^, two orders of magnitude higher than that reported for the methylammonium lead iodide (CH_3_NH_3_PbI_3_) embedded into poly(vinylfluoride(PVDF) nanofibers, reported to be 12 μWm^−2^ for R*_l_* = 10 MΩ [30]. Furthermore, our MDABCO-NH_4_I_3_@PVC piezoelectric generator is capable of delivering a peak power density with a magnitude similar to that achieved for electrically poled MDABCO-NH_4_I_3_ films deposited on a polyimide substrate after a preheating treatment up to 140 °C. Here, the piezoelectric generator delivered a peak power density of 2000 μWm^−2^ under an R*_l_* = 250 MΩ [48]. Therefore, our electrospun-doped fiber mat can achieve a high peak power density without the need for electrical polling or previous heating treatment, which is very advantageous.

In the present work, we demonstrate that incorporating the organic lead-free perovskite MDABCO-NH_4_I_3_ into electrospun PVC fibers, processed at room temperature without poling, is an easy and straightforward way to fabricate piezoelectric generators using lead-free perovskite nanocrystals as active materials. Moreover, the piezoelectric voltage coefficient is defined as
*g_eff_
*= *d_eff_*/(ε′ε_0_) VmN^−1^
which, as an important quantity for quantifying the performance of a material for integration as a piezoelectric sensor, was calculated for our electrospun fiber mats. For MDABCO-NH_4_I_3_@PVC, ε′= 50 at 20 Hz and *g_eff_
*= 3.6 VmN^−1^. This extremely high piezoelectric voltage coefficient is one order of magnitude higher than that displayed by a polyvinylidene fluoride (PVDF) polymer thin film for which *g_eff_* = 0.29 VmN^−1^ [49,50] and six times higher than that exhibited by the layered lead perovskite (4-aminotetrahydropyran)_2_PbBr_4_, which was reported to be *g_eff_
*= 0.67 VmN^−1^ [11].

## 4. Conclusions

In this study, we show that the lead-free organic ferroelectric perovskite N-methyl-N′-diazabicyclo[2.2.2]octonium)–ammonium triiodide (MDABCO-NH_4_I_3_) nanocrystals incorporated into electrospun fibers of three different polymers, processed at room temperature and without poling, can generate output voltages ranging from 2 V to ~17 V, using lead-free perovskite nanocrystals as active piezoelectric materials. In particular, we show that MDABCO-NH_4_I_3_ embedded in PVC fibers displays an effective piezoelectric voltage coefficient as high as *g_eff_* =3.6 VmN^−1^.

A piezoelectric nanogenerator (PENG) fabricated using an MDABCO-NH_4_I_3_@PVC electrospun fiber mat as the active piezoelectric component is demonstrated as a proof-of-concept. In addition, electrospun fibers exhibit intense blue photoluminescence at 325 nm, emitted by the embedded perovskite nanocrystals in the UV–vis–NIR and optical spectra. It is remarkable that MDABCO-NH_4_I_3_ inside the fibers show intense luminescence, which has not been reported before for this perovskite. Importantly, the pyroelectric coefficient of MDABCO-NH_4_I_3_ nanocrystals increases to the very high value of 194 × 10^−6^ Cm^−2^k^−1^ achieved at the ferroelectric–paraelectric transition. This pyroelectric coefficient has a magnitude within the same order as that reported for a semiorganic ferroelectric triglycine sulfate (TGS) single crystal. Additionally, the nanocrystals maintain their optical, piezoelectric, and pyroelectric properties for long periods when embedded into electrospun fibers, inhibiting perovskite oxidation, which is promoted by the polymer matrix that acts as a shielding.

## Figures and Tables

**Figure 1 materials-15-08397-f001:**
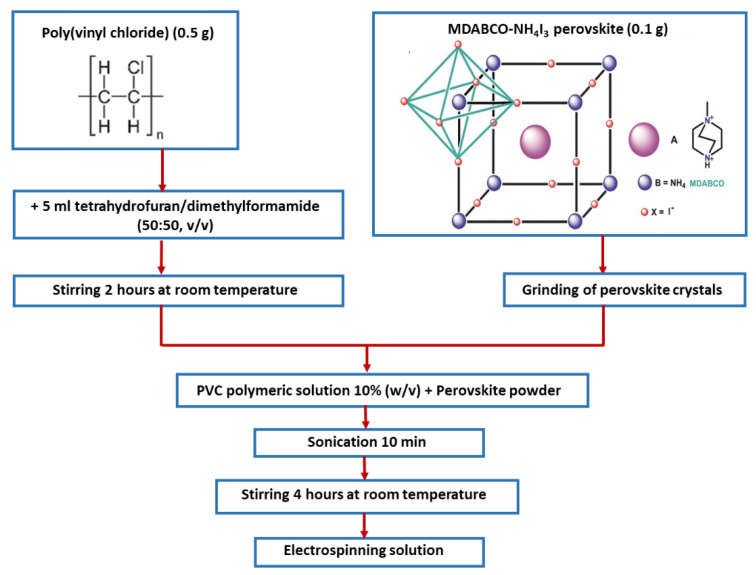
MDABCO-NH_4_I_3_ perovskite crystal flow chart for the preparation of MDABCO-NH_4_I_3_ @PVC electrospinning solution.

**Figure 2 materials-15-08397-f002:**
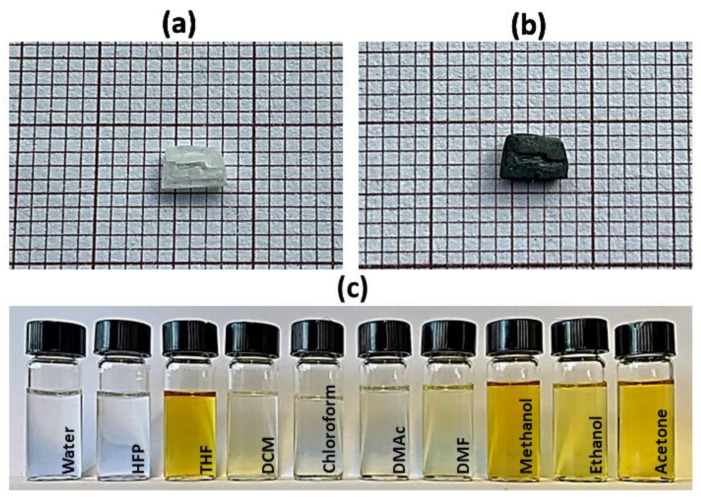
MDABCO-NH_4_I_3_ perovskite crystal, after several hours (**a**) and after 2 weeks (**b**) of exposure in the open air at room temperature. Perovskite solutions in different solvents (**c**), from left to right: water, hexafluoroisopropanol (HFP), tetrahydrofuran (THF), dichloromethane (DCM), chloroform, dimethylacetamide (DMAc), dimethylformamide (DMF), methanol, ethanol, and acetone.

**Figure 3 materials-15-08397-f003:**
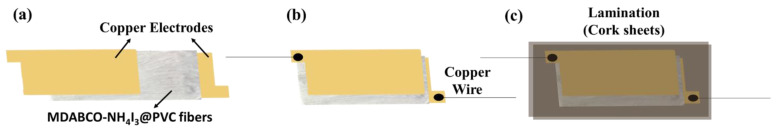
Schematic of MDABCO-NH_4_I_3_@PVC. (**a**) Electrospun nanofiber mat sandwiched between two copper electrodes. (**b**) Wires attached to the electrodes. (**c**) The complete laminated system with thin cork sheets.

**Figure 4 materials-15-08397-f004:**
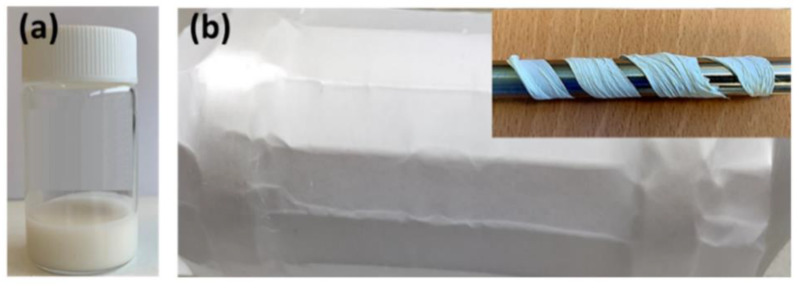
Perovskite polymeric solution (**a**) and MDABCO-NH_4_I_3_@PVC electrospun nanofiber mat (**b**) the inset shows a fiber mat folded around a cylindrical stick, demonstrating the flexibility of the fibers.

**Figure 5 materials-15-08397-f005:**
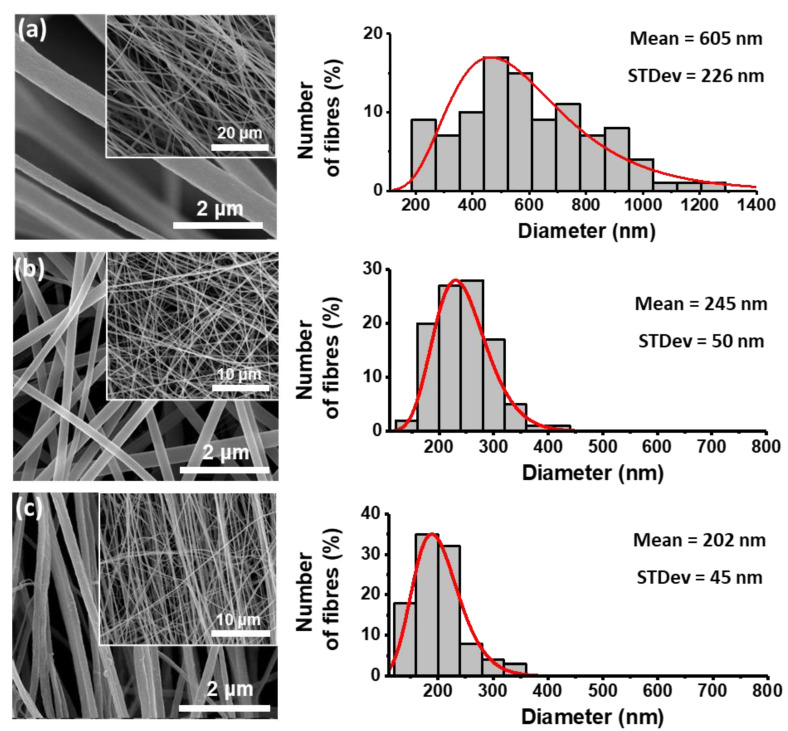
SEM images at magnification levels of 10,000× and 50,000× and respective fiber diameter distribution histograms for PMMA (**a**), PA66 (**b**), and PVC (**c**) nanofibers with embedded MDABCO-NH_4_I_3_ perovskite crystals. The red curves indicate logarithmic normal distributions using the mean and standard deviations of each set of fibers.

**Figure 6 materials-15-08397-f006:**
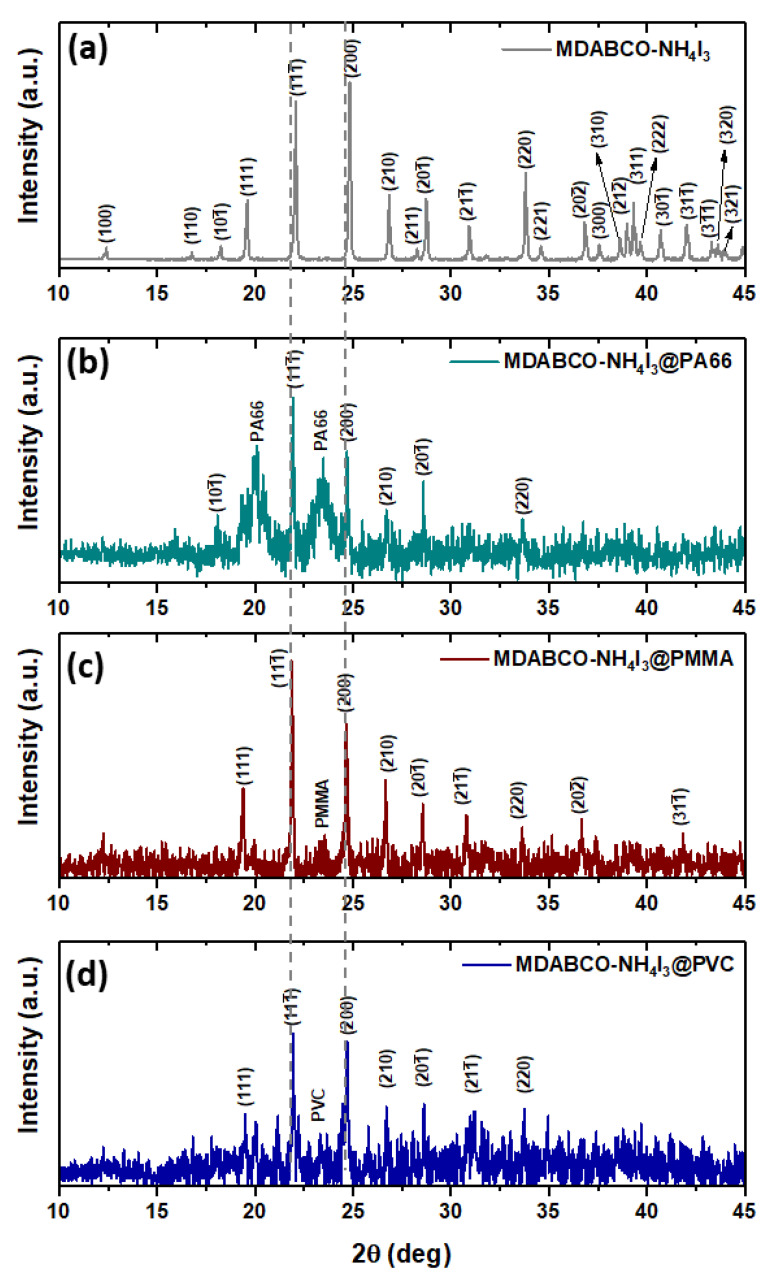
(**a**) Synthesized perovskite crystals with all Bragg peaks indexed from the published CIF file 1,836,322 [17]. X-ray patterns of PA66 (**b**), PMMA (**c**)**,** and PVC (**d**) nanofibers with embedded MDABCO-NH_4_I_3_ perovskite crystals between 10° and 45°.

**Figure 7 materials-15-08397-f007:**
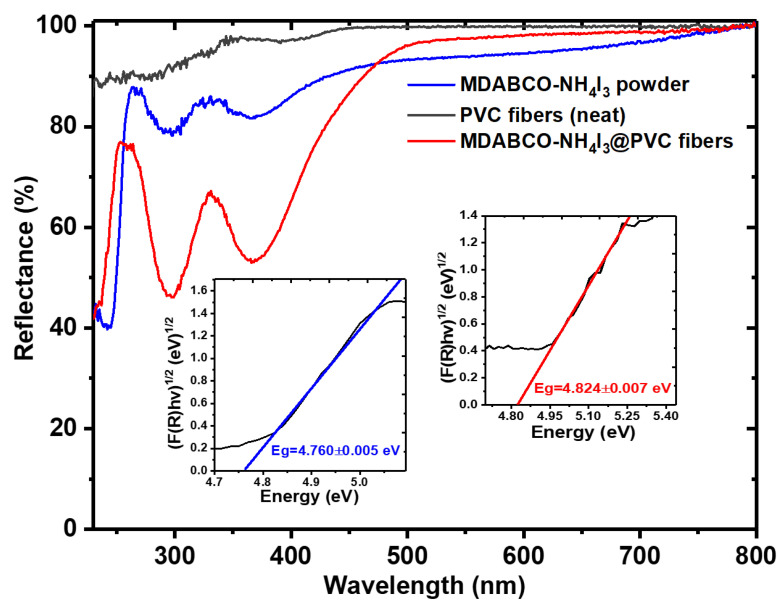
UV–vis reflectance of MDABCO-NH_4_I_3_ powder and electrospun PVC fibers with MDABCO-NH_4_I_3_ nanocrystals. The inset shows the Kubelka–Munk function indicating a band gap energy of 4.760 and 4.824 eV for powder and fibers, respectively.

**Figure 8 materials-15-08397-f008:**
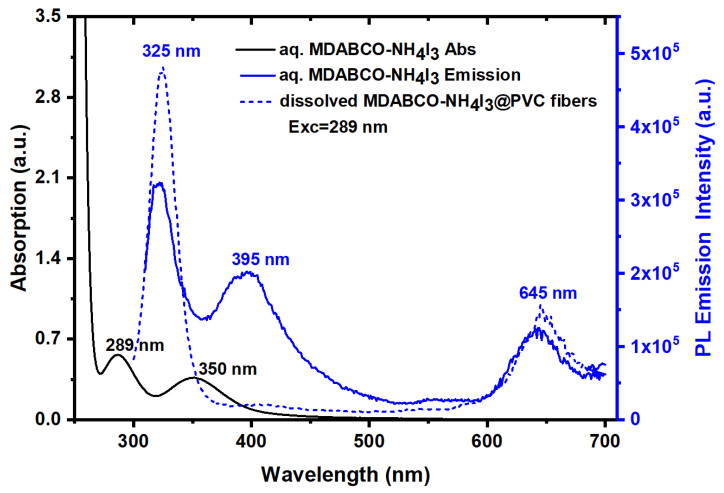
Optical absorption and photoluminescence emission of MDABCO-NH_4_I_3_ aqueous solution (3 mg/mL) and MDABCO-NH_4_I_3_@PVC nanofibers dissolved in tetrahydrofuran. The excitation wavelength for PL measurements were 289 nm.

**Figure 9 materials-15-08397-f009:**
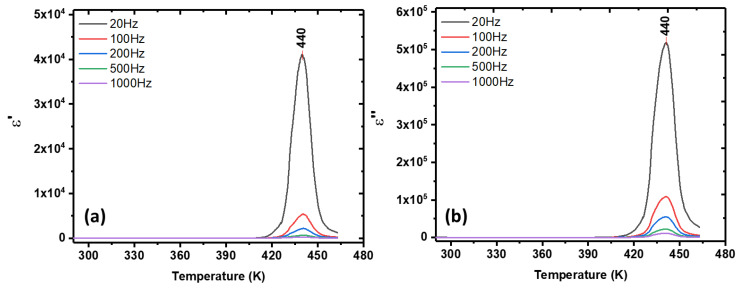
Dielectric permittivity of an MDABCO-NH_4_I_3_ polycrystalline sample showing its (**a**) real and (**b**) imaginary parts as functions of temperature and frequency. The ferroelectric–paraelectric phase transition occurs at 440 K.

**Figure 10 materials-15-08397-f010:**
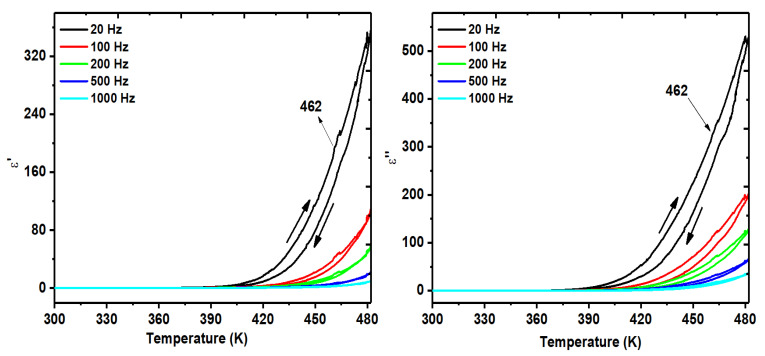
Dielectric permittivity of MDABCO-NH_4_I_3_ nanocrystals embedded in electrospun fibers, MDABCO-NH_4_I_3_@PA66a showing the ferroelectric–paraelectric phase transition at 462 K. In this figure, the up and down black arrows indicate respectively the heating and cooling cycles.

**Figure 11 materials-15-08397-f011:**
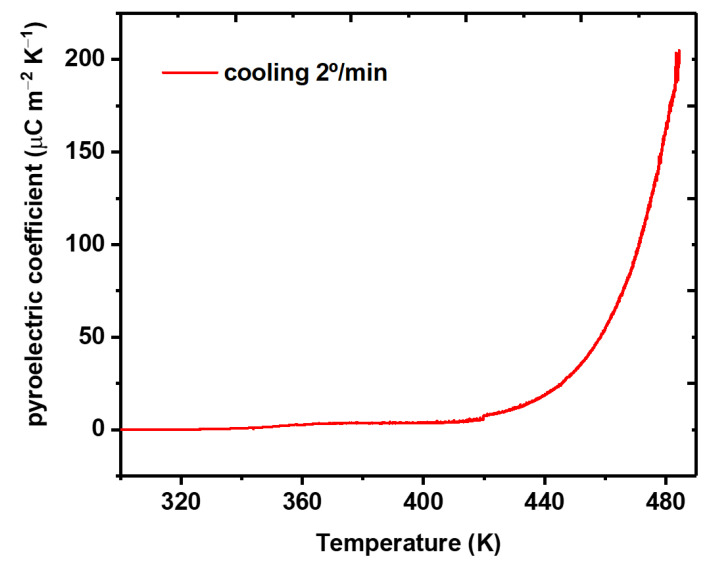
Pyroelectric coefficient as a function of the temperature of MDABCO-NH_4_I_3_@PA66 nanofibers, measured on cooling from the Curie transition temperature.

**Figure 12 materials-15-08397-f012:**
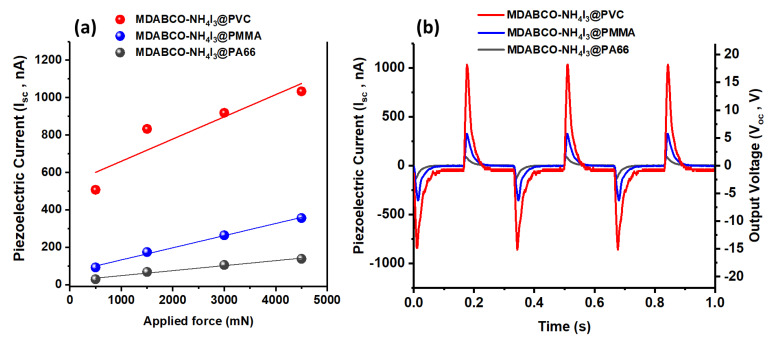
(**a**) Piezoelectric current as a function of the applied forces and (**b**) Output voltage and current as a function of time from MDABCO-NH_4_I_3_ incorporated into different electrospun polymer nanofibers.

**Table 1 materials-15-08397-t001:** The average crystallite size of MDABCO-NH_4_I_3_ perovskite crystals embedded in electrospun fibers.

MDABCO-NH_4_I_3_ in	Size (nm)
PVC	62 ± 12
PMMA	77 ± 06
PA66	83 ± 17

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
