# Peer review of "Lead-Free MDABCO-NH4I3 Perovskite Crystals Embedded in Electrospun Nanofibers"

_materials, 2022, doi:10.3390/ma15238397_

Round 1

Reviewer 1 Report

This is an interesting manuscript describing the pyroelectric and piezoelectric voltage coefficients of blue luminescent organic lead-free perovskite N-methyl-N'-diazabicyclo[2.2.2]octonium)–ammonium triiodide MDABCO-NH4I3 electrospun nanofibers. These polymer fibers are fabricated using electrospinning technique. This work is interesting. The authors are suggested to perform major revisions of the manuscript on the following points given below:

The comments are as follows:

(1)  The title should be “Lead free MDABCO-NH4I3 perovskite crystals embedded in electro spun nanofibers”.

(2)  What are the advantages of organic ferroelectric MDABCO-NH4I3 perovskite crystals for piezoelectric nanogenerator?

(3)  The authors should perform the switching polarity test to verify the purity of piezoelectric effects in the PENG device.

(4)  Authors should perform the stability test of the champion device.

(5)  Authors are suggested to add optical absorption and photoluminescence emission of MDABCO-NH4I3@PA66, MDABCO-NH4I3@PMMA nanofibers dissolved in tetrahydrofuran.

(6)  Why MDABCO-NH4I3@PVC nanofibers shows highest piezoelectric current and piezoelectric voltage?

(7)  Why the effective piezoelectric coefficient of MDABCO-NH4I3@PA66 is low?

(8)  Can the authors explain why the current output remains unchanged for MDABCO-NH4I3@PVC nanofiber mat for frequencies between 1Hz and 12Hz?

Author Response

Response to Reviewer 1

We thank you very much for your helpful comments and suggestions, which we address below.

1.The title should be “Lead free MDABCO-NH4I3 perovskite crystals embedded in electro spun nanofibers”.

Authors Response: The title was changed to that proposed.

  1. What are the advantages of organic ferroelectric MDABCO-NH4I3 perovskite crystals for piezoelectric nanogenerator?

Authors Response: The main advantage of organic ferroelectric MDABCO-NH4I3 perovskite crystals for piezoelectric nanogenerators when embedded into electrospun fibers, results both from their increased stability inside the polymer matrix accompanied by their high output piezoelectric voltage. Perovskite compounds with halogens atoms, in particular Iodine, are unstable in open air at room temperature as discussed in the manuscript. By replacing Pb and other toxic ions by organic cations non toxic metal- free materials can be designed. Therefore, organic materials have become an attractive area research topic due to their remarkable structural variability and highly tunable properties.

  1. The authors should perform the switching polarity test to verify the purity of piezoelectric effects in the PENG device.

Authors Response: The switching polarity test was included in Supplementary Information, Figure SI7.

  1. Authors should perform the stability test of the champion device.

Authors Response:

The stability test was included in Supplementary information, Figure SI8 c).

In the manuscript at line 382, the following test was added:

“Figure SI7, shows the output voltage generated from MDABCO-NH4I3@PVC nanofiber mat with reverse polarity. To analyze the reproducible behavior of the piezoelectric active nanofiber mat as a nanogenerator, a stability test was performed during a time interval of 4 h, uninterruptedly, under a periodical force applied with 2.7 N at a frequency of 3 Hz, Figure SI8. The nanogenerator output voltage does not decrease overtime. This is an important property indicating that the MDABCO-NH4I3 perovskite nanocrystals may be used to integrate future nanogenerators devices.”

  1. Authors are suggested to add optical absorption and photoluminescence emission of MDABCO-NH4I3@PA66, MDABCO-NH4I3@PMMA nanofibers dissolved in tetrahydrofuran.

Authors Response: The polymers PA66 and PMMA are not soluble in tetrahydrofuran. Besides any of the polymers used are luminescent. As Figure 8 in the manuscript shows, the emission is only due to MDABCO-NH4I3 compound.

  1. Why MDABCO-NH4I3@PVC nanofibers shows highest piezoelectric current and piezoelectric voltage?

Authors Response: As written in the manuscript, there is a contribution from PVC polymer which is itself piezoelectric.

  1. Why the effective piezoelectric coefficient of MDABCO-NH4I3@PA66 is low?

Authors Response:  We think that the smaller value of the effective piezoelectric coefficient of MDABCO-NH4I3@PA66, might result from the fact that the perovskite nanocrystals size is the highest for PA66 polymer fibers. The influence of crystal size on the piezoelectric output properties is a subject, which we will certainly address in another study in the very near future.

  1. Can the authors explain why the current output remains unchanged for MDABCO-NH4I3@PVC nanofiber mat for frequencies between 1Hz and 12Hz?

Authors Response: The output current remains unchanged with the frequency variation because there is no externally applied AC electric field. The response from the material current results only from the internal material polarization developed due to an applied periodic stress. One should note that the frequency of the applied force, in the measured frequency region of 1-10 Hz, is a low frequency region where the nanoenergy harvesting applications are of the interest like happens for human body movements as an example.

Reviewer 2 Report

In the present work, authors reported the synthesis of three different polymer fibers embedded MDABCO-NH4I3 perovskite crystals by using electrospinning technique, and then investigated their structural, luminescence, pyroelectric and piezoelectric properties. Results indicated that the nanofibers behave as active piezoelectric energy harvesting sources that produce a piezoelectric voltage coefficient up to geff = 3.6 VmN-1 and show a blue intense luminescence band at 325 nm. Overall, this work has certain function reference as an applied research. However, some issues should be addressed.

1, In Abstract section, authors should figure out the significance and real-life application of the present paper.

2, The introduction writing part need to be improved. Also, the writing and presentation of the introduction lacks a bit in clarity. The paper requires some amount of rewriting to clarify all aspects of it, especially the novelty and new findings of this work that need to be clearly mentioned. In addition, authors may remove summary of the results from the final paragraph of the introduction section.

3, If there is process flow diagram can be added in fig. 1, it would be helpful to non-specialist readers. Additionally, how about the particle distribution during mixing process and the varied ratio of it after the electrospinning process?

4, High crystallinity is a determinant in applications of nanoparticles. How to adjust the crystallinity of particles in this work? The authors should also pay attention to this challenge, and some pioneering and original researches about controllable modulation of crystallinity are highly suggested: ACS Applied Materials & Interfaces, 2017, 9, 16404; Giant, 2021, 8, 100076; Angew. Chem. Int. Ed., 2015, 54, 4571.

5, In the present work, it might provide better insights about the influence of defect structures of perovskite on properties. The discussions correlating the optical absorption and photoluminescence, and the defects should be deepened for clarity.

6, It was claimed that “This results from the fact that the nanocrystals are immersed in the polymer matrix, making it necessary to go higher in temperature for the dispersed nanocrystals to make the ferroelectric-paraelectric transition temperature”. So why? Authors may explain it or refer to the previous work.

7, As we all know that the size of nanoparticles plays an important role in influencing dielectric performance. But it only said that “It also indicates a diffuse character of the phase transition, induced by the small size of the MDABCO NH4I3 nanocrystals, embedded in the polymer matrix.”. Please discuss the influence of size of nanoparticles on dielectric in deep.

Author Response

Response to Reviewer 2

We thank you very much for your helpful comments and suggestions. They are address below.

  1. In Abstract section, authors should figure out the significance and real-life application of the present paper.

Authors response: The abstract was modified in order to clearly demonstrate the significance of the present work. It is now:

Abstract: In this work we introduce lead-free organic ferroelectric perovskite N-methyl-N'-diazabicyclo[2.2.2]octonium)–ammonium triiodide, MDABCO-NH4I3, nanocrystals embedded in three different polymer fibers fabricated by the electrospinning technique, as mechanical energy harvesters. Molecular ferroelectrics offer the advantage of structural diversity and tunability, easy fabrication and mechanical flexibility. Organic-inorganic hybrid materials are new low symmetry emerging materials which may be used as energy harvesters due to their or piezoelectric or ferroelectric properties. Among these, ferroelectric metal free perovskites are a class of recently discovered multifunctional materials. The doped nanofibers, which are very flexible and have a high Young modulus, behave as active piezoelectric energy harvesting sources that produce a piezoelectric voltage coefficient up to geff = 3.6 VmN-1 and show a blue intense luminescence band at 325 nm. In this work, the pyroelectric coefficient is reported for the MDABCO-NH4I3 perovskite inserted in electrospun fibers. At the ferroelectric-paraelectric phase transition, the embedded nanocrystals display a pyroelectric coefficient as high as 194×10-6 Cm-2k-1, within the same order of magnitude as that reported for the state-of-the-art bulk ferroelectric triglycine sulfate (TGS). The perovskite nanocrystals embedded into the polymer fibers remain stable in their piezoelectric output response, and no degradation is caused by oxidation, making the piezoelectric perovskite nanofibers suitable to be used as flexible energy harvesters.

  1. The introduction writing part need to be improved. Also, the writing and presentation of the introduction lacks a bit in clarity. The paper requires some amount of rewriting to clarify all aspects of it, especially the novelty and new findings of this work that need to be clearly mentioned. In addition, authors may remove summary of the results from the final paragraph of the introduction section.

Authors response:  The introduction has been rewritten for clarification. The summary at the end was removed. The parts modified are shown in yellow:

Mechanical energy harvesting at low frequencies, from materials that are environmentally friendly and scavenging energy from multiple sources as for example the human body movements are at the forefront of research [1,2].

Ferroelectrics are inherently piezoelectric and pyroelectric materials; that is, they are able to produce an intrinsic electrical potential difference in response to an applied force (or originate a mechanical movement due to an applied electric field) and a temperature gradient, respectively.

Valasek discovered the first ferroelectric, Rochelle salt or potassium sodium tartrate tetrahydrate [KNaC4H4O6] (4H2O), in 1920 and was in fact the first semi-organic molecular ferroelectric crystal that is also non-toxic [3,4].

Among ferroelectrics, inorganic perovskites, formula ABX3 (A, B = metal cations, X = anion; usually an oxide), are a well-known family of solid-state inorganic compounds finding applications like capacitors, sensors, actuators, etc. The best known are metal oxides such as strontium, barium, or lead titanate (SrTiO3, BaTiO3, PbTiO3, respectively), their solid solutions such as Pb(Zr,Ti)O3 (PZT), niobates such as PZN (PbZn1/3Nb2/3O3) (PZN), (PbMg1/3Nb2/3O3) (PMN) and lithium niobate LiNbO3). These materials have, until now, been used largely by several industries due to their functional properties, combining ferroelectricity with nonlinear optical and electro-optic effects as well as multiferroicity [5,6].

So far, the commercially available piezoelectrics are dominated by inorganic per-ovskites, namely PZT-based materials, and polymers such as polyvinylidene difluoride (PVDF) and its modifications such as PVDF-TrF [7,8]. However, as a result of lead toxicity, lead-based ferroelectrics are presently a serious environmental hazard. These concerns originated active research on substituting those perovskite-type materials, one ion A or X, with a molecular building unit [9,10].

Hybrid organic-inorganic perovskites (HOIPs) are a recent class of ferroelectric crystalline materials for optoelectronic applications, which are competitive with the inorganic perovskites. These semi-organic ferroelectrics possess many advantages when compared with inorganic ones. For example, they can be synthesized at room temperature, they are more flexible and with lower weight than their inorganic counterparts, and they have remarkable structural variability resulting in high tuneable properties. Therefore, they became an attractive topic of research towards their application as piezoelectric and pyroelectric materials that replace inorganic materials [11-14]. Importantly, highly efficient solar cells have been demonstrated using methylammonium lead halide perovskites, which enabled the search for lead free perovskites. Lead-free HOIPs are a recently discovered and highly promising family of perovskites [15-20].

A lead free organic-inorganic perovskite recently discovered is (N-methyl-N'-diazabicyclo[2.2.2]octonium)–ammonium triiodide, MDABCO-NH4I3, which has a spontaneous polarization of 22 µC/cm2, close to that of barium titanate (which is around 26 µC/cm2), a high phase transition temperature at 448 K and several polarization directions. It displays attractive properties for applications in flexible optoelectronic devices [21,22].

The fabrication of structures at the nanoscale has been attracting increased amount of attention because of their size-dependent properties. One-dimensional structures such as nanowires, nanotubes, and nanofibers are the smallest dimensional structures displaying new properties with potential applications in fields such as electronics, photonics, sensing, and energy harvesting.

Electrospinning is a well-established technique for forming micro- and nanoscale fibers with a large surface-to-volume ratio forming mats of several square centimetres area. Electrospun fiber mats are nanostructured multifunctional materials drawn from a precursor polymeric solution blended with functional nanoparticles under very strong static electric fields [23-27].

In addition, the nanofiber anisotropic shape and large surface area ratio contribute to an increase in their mechanical strength and flexibility. In this context, nanoscale ferroelectrics with perovskite structure are a promising research area [11,28].

One application of functional electrospun fibers is in the harvesting of electrical nanoenergy at low frequencies through the piezoelectric effect, due to the polarization induced by the material deformation [29]. Piezoelectric nanogenerators, usually called PENGs, show potential for powering low-power devices. An example of the use of a semiorganic perovskite as a PENG was reported for the methylammonium lead iodide (CH3NH3PbI3) incorporated in PVDF polymer fibers made by electrospinning: an output voltage of approximately 220 mV at 4 Hz, under an applied force of approximately 7.5 N, a maximum output power of 0.8 mW/m2 was generated [30].

In this manuscript, MDABCO-NH4I3 perovskite embedded into electrospun nanofibers, are capable of acting as lead-free piezoelectric (PENG) nanogenerators for effective mechanical energy harvesting. In particular, for poly(vinyl chloride) (PVC) polymer, an instantaneous output power density of 2020 μWm-2 is delivered under the application of a mechanical periodical force. The pyroelectric coefficient of a polycrystalline MDABCO-NH4I3 in electrospun fibers, has a similar order of magnitude of that dis-played hybrid ferroelectric triglycine sulfate (TGS).

  1. If there is process flow diagram can be added in fig. 1, it would be helpful to non-specialist readers. Additionally, how about the particle distribution during mixing process and the varied ratio of it after the electrospinning process?

Author response:

A flow diagram was added and is now Figure 1 in the manuscript.

Figure 1. MDABCO-NH4I3 perovskite crystal flow chart for the preparing of MDABCO-NH4I3@PVC electrospinning solution.

  1. High crystallinity is a determinant in applications of nanoparticles. How to adjust the crystallinity of particles in this work? The authors should also pay attention to this challenge, and some pioneering and original researches about controllable modulation of crystallinity are highly suggested: ACS Applied Materials & Interfaces, 2017, 9, 16404; Giant, 2021, 8, 100076; Angew. Chem. Int. Ed., 2015, 54, 4571.

Author response:

We agree that crystallinity of nanoparticles is a very important subject on its own. In our manuscript, we addressed that question in the beginning of Results section. We demonstrated that MDABCO-NH4I3 perovskite nano crystals were indeed embedded into the polymer fibers and determined their average size. However, a study of the influence and control of crystallinity and crystallite size was not the focus of the present study. We intend in the near future, to perform a detailed study on the relationship between crystallite size and physical properties like piezoelectricity and nonlinear optical second harmonic generation.

  1. In the present work, it might provide better insights about the influence of defect structures of perovskite on properties. The discussions correlating the optical absorption and photoluminescence, and the defects should be deepened for clarity.

Author response:

In section 3.3 of the manuscript, the linear optical properties are discussed but we did not mention that they were related to defects in the perovskite crystal structure. Figure 7 shows PL emission from MDABCO-NH4Inanocrystals embedded into the fibers and it was not attributed to any defects.  The first published paper about MDABCO-NH4I3, reference 21 in our manuscript, does not also consider its photoluminescence as being due to structural defects.

  1. It was claimed that “These results from the fact that the nanocrystals are immersed in the polymer matrix, making it necessary to go higher in temperature for the dispersed nanocrystals to make the ferroelectric-paraelectric transition temperature”. So why? Authors may explain it or refer to the previous work.

Author response:

When ferroelectric nanostructures are under stress/strain, their ferroelectric transition temperature varies, as compared to bulk unstressed ones. The shift in transition temperature depends on the strain state of the crystal. For example, hydrostatic strain tends to decrease the transition temperature, while anisotropic strain states can strongly increase the transition temperature [J. Belhadi, M. El Marssi, Y. Gagou, Yu. I. Yuzyuk, and I. P. Raevski, Giant Increase of Ferroelectric Phase Transition Temperature in Highly Strained Ferroelectric [BaTiO 3 ] 0.7Λ /[BaZrO 3 ] 0.3Λ Superlattice, EPL Europhys. Lett. 106, 17004 (2014)]. In our case, from XRD we have observed a slight shift of the nanocrystals XRD peaks indicating that the nanocrystals are under strain. This nanofiber induced strain on the nanocrystals is anisotropic, due to the high aspect ratio of the nanofibers, and has increased the transition temperature as compared to the bulk. As such, in order to clarify this, we have introduced 3 additional references dealing with these matters and have included the following sentence in the manuscript:

When ferroelectric nanostructures are under stress/strain, their ferroelectric transition temperature varies, as compared to bulk unstressed ones. The shift in transition temperature depends on the strain state of the crystal. For example, hydrostatic strain tends to decrease the transition temperature [38], while anisotropic strain states can strongly increase the transition temperature [39,40]. In our case, from the X-ray diffraction results of figure 6, we observe a slight shift of the XRD peaks as compared to the bulk, indicating the MDABCO-NH4I3 nanocrystals are under strain inside the fibers. This nanofiber induced strain is anisotropic, due to the high aspect ratio of the nanofibers, and has increased the transition temperature as compared to the bulk. As such, the observed increased transtion temperature results from the fact that the nanocrystals are immersed in the polymer matrix, making it necessary to go higher in temperature for the dispersed nanocrystals to make the ferroelectric-paraelectric transition temperature.

  1. As we all know that the size of nanoparticles plays an important role in influencing dielectric performance. But it only said that “It also indicates a diffuse character of the phase transition, induced by the small size of the MDABCO NH4I3 nanocrystals, embedded in the polymer matrix.”. Please discuss the influence of size of nanoparticles on dielectric in deep.

Author response:

We agree with the referee that the grain size influences the dielectric properties of the materials. Results from the literature indicate that, as compared to the bulk, the reduction of the nanoparticles sizes tend to decrease the permittivity and widen its transition temperature region, becoming a more diffuse transition [Ferroelectrics, 400:117–134, 2010, Journal of the European Ceramic Society 34 (2014) 2933–2939]. As such, in order to better support the assertion regarding the nanoparticles diffuse transition character we change the sentence, as shown below, and have included two additional references that review these aspects.

The permittivity results also indicate a diffuse character of the phase transition, widened as compared to the bulk, induced by the small size of the MDABCO-NH4I3 nanocrystals [41,42], embedded in the polymer matrix.

Round 2

Reviewer 1 Report

Revision completer. Accept

Reviewer 2 Report

Issues were addressed, and this work can be accepted.